# Wheat-Bran-Based Artificial Diet for Mass Culturing of the Fall Armyworm, *Spodoptera frugiperda* Smith (Lepidoptera: Noctuidae)

**DOI:** 10.3390/insects13121177

**Published:** 2022-12-19

**Authors:** Shishuai Ge, Bo Chu, Wei He, Shan Jiang, Chunyang Lv, Lingyun Gao, Xiaoting Sun, Xianming Yang, Kongming Wu

**Affiliations:** 1State Key Laboratory of Ecological Pest Control for Fujian and Taiwan Crops, Fujian Agriculture and Forestry University, Fuzhou 350002, China; 2State Key Laboratory for Biology of Plant Diseases and Insect Pests, Institute of Plant Protection, Chinese Academy of Agricultural Sciences, Beijing 100193, China; 3College of Plant Protection, Henan Agricultural University, Zhengzhou 450002, China; 4Key Laboratory of Economic and Applied Entomology of Liaoning Province, College of Plant Protection, Shenyang Agricultural University, Shenyang 110866, China; 5College of Tropical Crops, Hainan University, Haikou 570228, China

**Keywords:** fall armyworm, artificial diet, mass rearing, developmental period, flight ability

## Abstract

**Simple Summary:**

The fall armyworm (FAW), *Spodoptera frugiperda* Smith (Lepidoptera: Noctuidae), has become a global agricultural pest. Reducing the rearing cost of FAWs will be helpful for developing new environmentally friendly management strategies for this pest. As such, we researched an artificial wheat-bran-based diet for FAWs. The biological indicators (developmental duration, pupal weight, reproductive parameters, and flight ability) of the FAWs fed this diet were consistent with those of FAWs fed a traditional artificial diet in a laboratory setting. The rearing cost of using this new diet was lower by nearly 24%. These findings indicate that this artificial diet, based on wheat bran, is suitable for rearing FAW larvae, which will effectively reduce the rearing cost of FAWs.

**Abstract:**

*Spodoptera frugiperda* Smith (fall armyworm (FAW)) has invaded many countries in Africa and Asia in recent years, considerably restricting global agricultural production. In this study, we assessed the rearing performance of four artificial diets (D_1_: an artificial FAW diet based on wheat bran and soybean, maize, and yeast powders; D_2_: an artificial diet developed for *Helicoverpa armigera* (Hübner), based on wheat bran and soybean and yeast powders; D_3_: an artificial diet based on soybean powder; D_4_: an artificial diet based on wheat bran) for FAWs. We designed D_4_ based on a traditional diet (D_2_) but substituted the wheat bran for soybean and yeast powders. At 25 ± 1 °C, 75% ± 5% RH, and a 16:8 h L:D photoperiod, the larval stage of FAWs fed on D_4_ lasted 15.88 d, the pupal stage lasted 9.48 d, the pupal mass was 270.45 mg, the number of eggs deposited was 1364.78, and the mating rate was 89.53%. Most biological indicators of the larvae that were fed D_4_ were basically consistent with those of the larvae fed on the traditional diet (D_2_), but the intrinsic rate of increase (*r*), finite rate of increase (*λ*), and net reproduction rate (*R*_0_) of the D_4_ FAWs were lower than those of the D_2_ FAWs. The flight capacity (flight distance, duration, and velocity were 19.73 km, 6.91 h, and 2.90 km/h, respectively) of the D_4_ FAWs was comparable to that of the FAWs fed a traditional diet and maize leaves. Compared with the three other formulas, the cost of using D_4_ was lower by 26.42% on average. These results show that using cheap wheat bran instead of soybean flour and yeast powder as the basic material for an artificial diet for FAWs is feasible, which will substantially reduce rearing costs and promote the development of new controlling measures for FAWs. In addition, this study also has a reference value for reducing the cost of artificial diets for other insects.

## 1. Introduction

*Spodoptera frugiperda* (J.E. Smith) (fall armyworm (FAW)) originates from the tropical and subtropical regions of the Americas [1] and has recently developed into an important global agricultural pest [2,3,4,5]. FAWs are polyphagous herbivores [1,6]; the larvae can feed on a variety of important food and cash crops, such as maize, wheat, rice, cotton, sugarcane, and soybean. The invasive populations of FAWs are mainly maize-strain specific [7]: the preferred host is maize, and the seedlings and tender parts of maize plants are the preferred parts. Pest attacks can reduce maize production levels by 12% [8] and by over 40% in severely affected areas [9].

Chemical control strategies are still the main measures used against FAWs in many areas, especially in newly invaded areas (such as Africa and south and southeast Asia). The long-term use of chemical pesticides leads to a series of problems, such as environmental pollution, the killing of natural enemies, and enhanced pest resistance [10]. Studies have shown that FAW populations have developed high resistance to dozens of insecticides, e.g., carbamate, organophosphates, and pyrethroids [7,11,12]. Moreover, due to the long-term, large-scale planting of transgenic insect-resistant maize in the Americas, FAWs have also developed a certain resistance to most transgenic maize varieties [13,14,15]. For example, Cry1F maize was first commercially planted in Brazil in 2009, and FAWs became resistant to it within three years [16].

The long-term use of chemical pesticides or planting a single resistant crop variety does not conform to the ideal of integrated pest management (IPM). Sterile insect technology (SIT) is an environmentally friendly strategy with considerable potential [17]. Since the 1950s, SIT has been widely used to control a variety of agricultural pests and threats to human health [18,19,20]. SIT has been used to control the lepidopteran pests *Pectinophora gossypiella* (Saunders) and *Cydia pomonella* (L.) in the United States and Canada, respectively, achieving effective results [21,22]. We recently evaluated the effects of X-ray irradiation on FAWs and determined the appropriate doses for sterilization [23]. After irradiation with a dose of 250 Gy, the sterility rate was more than 85%, dramatically limiting the number of offspring. We thus theoretically confirmed the feasibility of using SIT via X-ray radiation for the control of FAWs in the field. Although the FAW is a typical migratory insect [24,25], due to the absence of diapause [1], this insect may mainly concentrate in maize fields as a year-round breeding ground, particularly in winter. Southwest China is a suitable wintering area for FAW, and sweet maize is one of the preferred hosts of FAW [26]. Sweet maize, as a fresh food, provides considerable economic benefits to the local area; thus, chemical pesticides and resistant plant varieties cannot be used to control local pests. Thus, we think the application of SIT in winter may be an effective method of controlling FAWs in their year-round breeding grounds. SIT requires a sufficient supply of FAWs for field release. In addition, the use of natural enemy insects and viral pesticides for pest control is rapidly developing [27,28,29] and producing these products will also require large quantities of FAWs.

In response to various research and economic needs, artificial diets for rearing insects in the laboratory have rapidly developed [30,31,32]. Given the economic importance of the FAW, especially with its worldwide spread, studies of artificial diets for FAWs have rapidly increased in recent years [33,34,35,36,37,38,39]. The basic materials used in FAW diets are soybean flour, wheat germ, and yeast powder [33,34,35,36,37,38,39]. The generally high price of these materials made rearing FAWs extremely expensive, which limited the development of various studies on FAWs. As such, in this study, we aimed to produce a cost-effective FAW artificial diet to reduce the rearing costs. We used two artificial and two traditional diets as follows: D_1_: an artificial diet based on soybean, maize, and yeast powders and wheat bran; D_2_: an artificial *Helicoverpa armigera* (Hübner) diet based on soybean and yeast powders and wheat bran; and two new diets designed based on D_2_, i.e., D_3_ (based on soybean powder) and D_4_ (based on wheat bran). We assessed the effects of these diets, using a diet of maize leaves as the control, on the development, survival, reproduction, and flight ability of FAWs using two sex life tables and a flight mill and evaluated the basic material costs and economic benefits of these diets.

## 2. Materials and Methods

### 2.1. Insects

In January 2019, we collected FAW larvae from maize fields in Dehong Prefecture (Yunnan, China). We transported the larvae to the laboratory and fed them maize leaves until pupation. After eclosion, we left the moths to lay eggs. We kept the offspring larvae in transparent plastic containers (22 × 15 × 8 cm; 30–40 individuals per container) and fed them an artificial diet (D_2_) [40], thus establishing a laboratory FAW colony. We reared both the larvae and adults under controlled conditions at 25 ± 1 °C, 70% ± 10% RH, and a 16:8 h L:D photoperiod. At the time of this study, the FAWs of the laboratory population had been reared over 20 generations with no abnormalities, which indicated that D_2_ fully met the long-term rearing needs of FAWs.

### 2.2. Diet Formula and Preparation Method

Both D_3_ and D_4_ were based on D_2_. We designed D_3_ by replacing all the wheat bran and yeast powder in D_2_ with soybean flour, and we designed D_4_ by replacing all the soybean flour and yeast powder in D_2_ with wheat bran. The specific formulas of the four diets are shown in Table 1.

We sterilized the wheat bran, soybean powder, and maize powder in a pressure cooker (120 °C, 30 min) before use. We weighed each substance according to the diet formulas provided in Table 1. According to the type of diet, we placed the wheat bran and soybean, yeast, and maize powders into a container and mixed them well. We heated 800 mL of distilled water (D_1_: 500 mL) to boiling point and added sorbic acid, waited for the sorbic acid to dissolve, then poured it into the container and stirred well. We heated 600 mL of distilled water, added agar and casein, continuously stirred until it boiled, and then poured the mixture into the container and stirred well. We dissolved ascorbic acid and vitamins into 100 mL of distilled water, which we added to the container when the temperature had decreased to approximately 40 °C. We then added formaldehyde and glacial acetic acid, stirred well, and poured the mixture into a sterilized box, which was cooled to room temperature and placed in a sealed refrigerator at 4 °C.

In addition to the four diets, we also used maize leaves to rear FAWs as the experimental control. We used the maize variety, Zhengdan 958 (Henan Qiule Seeds Technology, Co., Ltd., Zhengzhou, China). We cultivated the maize indoors under conditions of 26 ± 1 °C, 80% ± 5% RH, and a 16:8 h L:D photoperiod. We fed maize seedlings to the larvae at the 2–3-leaf stage.

### 2.3. Experimental Assays

#### 2.3.1. Effects of Different Diets on Development, Survival, and Reproduction of FAWs (*Spodoptera frugiperda*)

For the experiments, we reared FAWs in climate chambers (MGC- 450HP; Yiheng Technology Instrument Co., Ltd., Shanghai, China) with 5000 Lx illumination at 25 °C, 75% RH, and a 16:8 h L:D photoperiod. Each diet treatment consisted of 80 FAW individuals, with three replicates giving a total of 240 individuals. We collected egg masses deposited by adults from the laboratory colony at the same time and observed them daily until hatching. We placed the newly hatched larvae in plastic cups (25 mL, 38 × 30 × 30 mm) with an appropriate amount of diet or maize leaves. To prevent mechanical damage to or the death of larvae caused by human operation, we added 2–3 larvae to each cup, and we left one randomly selected healthy larva in the cup on the following day. We transferred the other larvae to a new cup (1 larva/cup), which we fed in the same way until the emergence and testing of their flight ability. We renewed the diets every 2–3 days (the maize leaves were changed daily) to avoid the influence of food freshness on the experimental results. We recorded the development and survival of the larvae daily until pupation. We determined the larval instar by the number of molts. The pupae were fragile at the early stage, so we weighed them on an electronic balance (0.0001 g, Mettler Toledo ME204; Beijing Haitian Youcheng Technology Co., Ltd., Beijing, China) on the third day after pupation. We checked the pupae for survival daily until death or eclosion. Following adult eclosion, we recorded the adult sex and checked the adults for deformity. We recorded the longevity of any deformed adult as 1 day. We paired healthy adults in disposable plastic cups (500 mL) (1 mating pair/cup). We covered each plastic cup with sterile gauze for collecting the egg masses and placed a cotton ball immersed in 10% *v*/*v* honey/water solution at the bottom of the cup to provide water and nutrients to the FAWs. On a daily basis, we refreshed the cotton ball and the gauze with egg masses. We replaced the cup if the egg masses were deposited on the cup wall or bottom. We documented the pre-oviposition period, daily fecundity, oviposition period, egg hatchability, and adult longevity. We determined the mating status via the presence of spermatophores in the female after death.

#### 2.3.2. Effects of Diet on Flight Ability of FAWs (*Spodoptera frugiperda*)

We transferred the adults in each diet treatment to different cages after eclosion and fed them a 10% *v*/*v* honey/water solution. We individually attached 2-day-old adults to a flight mill (FXMD-24-USB flight mill; Jiaduo Science, Industry and Trade Co., Ltd., Hebi, China) in artificial climate chambers using the method described by Ge et al. [41]. We gently collected active, undamaged adults from the rearing cage. We expanded their wings, brushed the scales from the junction of the abdomen and thorax, and performed tethering using a small droplet of cyanoacrylate glue (Deli Group Co., Ltd., Zhejiang, China). After the moths recovered, we attached only undamaged and active individuals to the flight mill, which was kept under 25 °C and 75% RH. The chamber was completely darkened, and the moths were allowed to engage freely in tethered flight. We conducted these tests from 8 p.m. until 8 a.m. the following day (Chinese standard time).

#### 2.3.3. Comparison of Cost of Different Diets

The purchase price of each material used in the diets and each vitamin are shown in Appendix A, respectively. According to Table 1, Appendix A, we calculated the weight and material cost of each diet. Then, we accordingly assessed the costs of culturing 1 million FAWs.

### 2.4. Data Analyses

To assess the effects of different diets (and maize leaves) on the development, reproductive parameters, and flight ability of FAWs, we used a one-way analysis of variance (ANOVA) followed by Tukey’s HSD post hoc test. Before ANOVA, we first tested the data for normality using the Shapiro–Wilk test and for homogeneity of variances using Levene’s test. If the data did not meet the conditions, we performed arcsine transformation; proportional data were arcsine square-root-transformed. We analyzed the interaction of diet and sex (with diet and sex as the fixed effects) on the development and pupal weight of FAWs using a general linear mixed model. We used log-rank (survival analysis) testing for differences and linear trends in the survival curves of FAWs fed different diets. We used SPSS Statistics for Windows, version 20.0 (IBM, Armonk, NY, USA) for the above analyses, with a significance level of *p* < 0.05. We used TWOSEX-MSChart to investigate the life tables (*l_x_*, *f_x_*, *m_x_*, and *l_x_m_x_*) and demographic parameters (*R*_0_, *T*, *r,* and *λ*) [42,43,44,45]. We used a paired bootstrap test with 100,000 replications in TWOSEX-MSChart for the precise estimation of the mean and standard error among the demographic parameters of FAWs fed different diets [43,46].

## 3. Results

### 3.1. Developmental Period and Reproductive Parameters

The diets considerably affected the developmental period of the FAWs at each life stage, except for the egg stage (Table 2). Specifically, the developmental duration of each instar phase significantly differed (first-instar larvae: *F*_4, 1195_ = 22.043, *p* < 0.001; second-instar larvae: *F*_4, 1195_ = 200.981, *p* < 0.001; third-instar larvae: *F*_4, 1190_ = 72.966, *p* < 0.001; fourth-instar larvae: *F*_4, 1188_ = 88.444, *p* < 0.001; fifth-instar larvae: *F*_4, 1183_ = 177.289, *p* < 0.001; sixth-/seventh-instar larvae: *F*_4, 1050_ = 160.393, *p* < 0.001), the larval stage (*F*_4, 1050_ = 504.817, *p* < 0.001), pupal stage (*F*_4, 901_ = 19.146, *p* < 0.001), egg-pupa stage (*F*_4, 901_ = 287.092, *p* < 0.001), adult stage (*F*_4, 901_ = 17.257, *p* < 0.001), and total generations (*F*_4, 901_ = 17.020, *p* < 0.001)) among the different diet treatments. FAWs that were fed maize leaves (14.22 d) had the shortest larval stages, followed by FAWs fed with D_2_ (15.54 d) and D_4_ (15.88 d). The larval stages in the D_1_ (18.90 d) and D_3_ (20.94 d) groups were significantly longer than those of FAWs fed on other diets, and some larvae developed into the seventh instar. The pupal stage of D_4_-fed FAWs was significantly shorter than that of FAWs fed the other three diets, but we found no significant difference between D_4_ and maize leaves. The adult stage of D_2_-fed FAWs was the longest (14.63 d), followed by that of those fed D_4_ (13.70 d), with no significant difference between them. The adult stage of D_3_-fed FAWs was the shortest, only 9.96 days. FAWs fed maize leaves had the shortest total generation (39.86 d); FAWs fed D_1_ had the longest total generation (44.70 d). We found significant differences among the different diet treatments in pupal mass (*F*_4, 901_ = 120.395, *p* < 0.001), with the highest in D_4_ (270.45 mg) and the smallest in the maize leaf group (209.17 mg).

The diets notably affected the reproduction of FAWs (Table 2). The pre-oviposition period (*F*_4, 353_ = 3.219, *p =* 0.013), oviposition period (*F*_4, 353_ = 10.608, *p* < 0.001), eggs deposited per female (*F*_4, 353_ = 17.466, *p* < 0.001), and mating rate (*F*_4, 10_ = 4.961, *p* = 0.018) significantly differed among the different diet treatments. FAWs fed on maize leaves (3.96 d) had the shortest pre-oviposition period, with no significant difference among the four artificial diets. The oviposition period and eggs deposited per female did not differ among the FAW groups fed D_2_, D_4_, and maize leaves and were significantly higher than those in D_1_ and D_4_. The mating rate of the D_3_-fed FAWs was significantly lower than that of those fed maize leaves, with no significant difference among the other diet treatments.

The larval stage (*F*_1_ = 5.087, *p =* 0.024), pupal stage (*F*_1_ = 273.503, *p* < 0.001), total generation (*F*_1_ = 27.84, *p* < 0.001), and pupal mass (*F*_1_= 26.376, *p* < 0.001) differed between the sexes among the different diet treatments, with no significant differences between the sexes in the adult stage (*F*_1_ = 2.906, *p* = 0.089). We noted significant diet × sex interactions in the pupal stage (*F*_4_ = 11.997, *p* < 0.001), adult stage (*F*_4_ = 3.477, *p* = 0.008), total generation (*F*_4_ = 4.305, *p* = 0.002), and pupal mass (*F*_4_ = 3.096, *p* = 0.015). The larval stage, pupal stage, adult stage (except D_3_ and D_4_), total generation, and pupal weight of females were smaller than those of males (Figure 1).

### 3.2. Survival Rate, Fecundity and Life Expectancy

Diet strongly affected the age-stage survival rate (*s_xj_*) of the studied *Spodoptera frugiperda* individuals (Figure 2). We identified significant differences in the larval survival rate among FAWs fed different diets (*F*_4, 10_ = 6.392, *p* = 0.008); the larval survival rate of D_3_-fed FAWs (71.25%) was significantly lower than that of the D_2_-fed FAWs (98.33%), with no significant difference among the other diet treatments. The pupal survival rate did not significantly differ among the diet treatments (*F*_4, 10_ = 1.669, *p* = 0.233). We found significant differences in the adult deformity rate of FAWs fed different diets (*F*_4, 10_ = 8.836, *p* = 0.003), with the highest adult deformity rate occurring in the group fed D_3_ (14.36%), followed by those fed with D_4_ (6.05%) and maize leaves (5.67%) (Table 3).

The results of the survival analysis indicated significant differences in the age-specific survival rate (*l_x_*) of the FAW groups fed different diets (*χ*^2^ = 63.168, *df* = 4, *p* < 0.001), with a significant linear trend (*χ*^2^ = 52.905, *df* = 1, *p* < 0.001) (Figure 3). The maximum value of age-specific fecundity of female adults (*f_x_*) fed D_1_, D_2_, D_3_, D_4_, and maize leaves was 132.16, 241.36, 130.90, 228.68, and 227.29, respectively. The age-specific fecundity (*m_x_*) and age-specific maternity (*l_x_m_x_*) curves indicated that the order of peak oviposition day was maize leaves > D_4_ = D_2_ > D_1_ > D_3_. The maximum values of *m_x_* of the FAW groups fed D_1_, D_2_, D_3_, D_4_, and maize leaves occurred on days 37, 33, 35, 33 and 30, respectively; and on days 37, 33, 42, 33, and 30 for *l_x_m_x_*, respectively. The age-stage specific life expectancy (*e_xj_*) curve showed that the life expectancy was longer for the FAWs fed D_1_ and D_3_, with the shortest *e_xj_* occurring for those fed D_2_ (Figure 4).

### 3.3. Life Table Parameters

Diet substantially affected the life table parameters of FAW (Table 4). The net reproductive rate (*R*_0_) of D_2_-fed FAWs was the highest (580.37 offspring/individual); therefore, this diet was the best for population growth, followed by D_4_ (410.23 offspring/individual) and maize leaves (384.64 offspring/individual), with the lowest *R_0_* for D_3_-fed FAWs (176.53 offspring/individual). The FAW populations fed on D_1_ and D_3_ had longer mean generation times (*T*) and lower finite rates of increase (λ), which indicates that D_1_ and D_3_ are not suitable for rapid FAW population growth.

### 3.4. Flight Ability

Over a 12-h assessment period, the flight ability of the FAW was significantly affected by diets (Table 5). The flight distance (*F*_4, 276_ = 2.891, *p* = 0.023) and velocity (*F*_4, 276_ = 2.902, *p* = 0.022) differed among different diet treatments, with no difference recorded for flight duration (*F*_4, 276_ = 2.055, *p* = 0.087). The flight ability of FAW in D_3_ was weak, and its flight distance and velocity were significantly lower than those in D_1_ and D_2_. The flight ability did not differ among other diet treatments.

### 3.5. Evaluation of Cost of Four Artificial Diets

We calculated the cost per ton of production for each diet (D_1_, D_2_, D_3_, and D_4_) as RMB 6890.00, 5670.00, 5900.00, and 3970.00, respectively. The total cost of materials per ton of wheat bran, soybean powder, and yeast powder for D_2_ was RMB 2041.37; the total cost per ton of wheat bran for D_4_ was only RMB 340.59. According to the rearing number of FAWs for each diet, the diet amounts of culturing 1 million FAWs for each diet (D_1_, D_2_, D_3_, and D_4_) were 6.62 t, 6.00 t, 5.03 t, and 6.54 t, respectively. Therefore, compared with D_2_, using D_4_ as the artificial diet for rearing 1 million FAWs will save RMB 8056.2 (about USD 1111.76) and reduce the cost by nearly 24% (Table 6).

## 4. Discussion

As a notorious global agricultural pest, the rearing cost for FAWs must be reduced, and controlling the cost of the artificial diet required is a key step towards this goal. In this study, we developed an inexpensive and high-performance wheat-bran-based artificial diet (D_4_) for FAWs. The biological indicators (e.g., development duration, pupal mass, reproductive parameters, and flight ability) of the FAWs fed this diet were consistent with those of the FAWs fed a traditional artificial diet (D_2_), and the rearing cost of the new diet was nearly 24% lower. These findings indicate that using cheap wheat bran instead of soybean and yeast powders as the base material of an artificial FAW diet is feasible, which will improve the cost-effectiveness of the mass rearing of FAWs.

Natural diets are limited by many factors (e.g., season, growth cycle, and conditions) and can easily rot and deteriorate, making the mass rearing of insects difficult. Artificial diets are advantageous over natural diets owing to their low cost and ease of obtaining, preserving, and use. The nutritional quality of artificial diets is generally evaluated by examining the biological indicators, e.g., the development, uniformity, survival rate, and fecundity of reared insects [47,48,49]. In this study, the development of FAWs fed the D_4_ diet was similar to that observed by Pinto et al., who reported a larval stage of 15.6 days, a larval survival rate of 92.0%, and a pupal mass of 253.3 mg [34]. According to Wang et al. [35] and Su et al. [36], the highest fecundity was 452 and 836 eggs/female, respectively; these values are significantly lower than that of the D_4_-fed FAWs (1364.78 eggs/female) in this study. The differences in these findings may be related to the nutritional quality of the diet, rearing conditions, and the study of insects. Laboratory populations of insects can become degraded after rearing multiple generations owing to rearing conditions or genetic factors [50,51]. Therefore, we will evaluate the rearing performance of D_4_-fed FAWs after multiple generations in long-term experiments.

An appropriate artificial diet should contain all the nutrients needed for insect development in appropriate proportions; otherwise, it will substantially affect the life activities of the insects [52,53,54]. For example, Wu et al. found that the protein content in an artificial diet enhanced the reproductive capacity of *H. armigera* within a certain range but inhibited it when this range was exceeded [55]. Soybean powder, yeast powder, and wheat germ are rich in protein and various nutrients, and they are usually used as the base materials of artificial diets. In this study, the soybean-powder-based diet (D_3_) exhibited poor effects in rearing FAWs, which may be related to an improper proportion of nutrients or the presence of protease inhibitors in this diet [56]. We also tried to replace all the wheat bran, soybean powder, and yeast powder in D_2_ with maize powder, but the reared larvae all died several days later. The FAW larvae that were fed D_1_ grew slowly, and some larvae developed into the seventh instar, which may have been related to the high proportion of maize powder in the diet. High maize powder content may inhibit the development of FAWs, in contrast to our previous assumptions. According to Chen et al., high contents of lysine, methionine, and tryptophan in maize may affect the development of some insects [57]. Thus, reducing the proportion of maize powder may improve the rearing effect of D_1_.

The use and nutritional value of wheat bran are often underestimated. Wheat bran is not only rich in nutrients but also has other features, such as good water retention, easy availability, and low cost [57]. Our study showed that an artificial diet for FAWs, using only wheat bran as the base material, was able to meet the nutritional requirements of this insect fully. According to Pascacio-Villafan et al., the high content and multiple sources of protein (yeast and wheat germ) in artificial diets did not improve the rearing quality of *Anastrepha ludens* (Loew) because these proteins could not be fully used by the larvae [58]. Reducing the yeast content in an artificial diet did not affect the rearing effect and substantially reduced the cost [59]. Combined with the results of this study, we speculate that some materials in many artificial diets may be unnecessary; therefore, cost reductions for other diets may also be possible. In addition to the high price of yeast powder, agar, and casein are also relatively expensive, accounting for more than half of the total material costs of D_2_ and D_4_. Reducing the amounts of agar and casein used or using cheaper alternatives will further reduce the cost. Researchers have previously reported several partial or complete alternatives to agar [60,61,62]. In addition, the cost of the mass rearing of FAWs can also be reduced by optimizing rearing equipment and methods.

In this study, we designed an economical and effective artificial diet for FAWs, which could substantially reduce the cost of the mass rearing of FAWs. FAW irradiation with X-rays after mass rearing for field release can be combined with other control methods, such as light trapping, pheromone trapping, and planting resistant crop varieties in year-round breeding regions, which will help to control the number of resident pests and migrants effectively, and achieve large-scale, sustainable, and long-term management of FAWs. The wintering area of FAWs involves multiple countries and regions, and the insect has cross-border migratory abilities [63]. Therefore, the implementation of SIT programs requires the joint efforts and collaboration of multiple affected countries or regions. In addition, the mass rearing of FAWs can be used for the propagation of natural enemy insects and the production of viral pesticides to promote the rapid development of biological control technology effectively. Much remains to be achieved to mass rear FAWs successfully. Overall, our findings can be applied to effectively reduce the rearing cost of FAWs, promote various studies of FAWs, and provide a reference for reducing the costs of artificial diets for other insects.

## 5. Conclusion

*Spodoptera frugiperda* (FAW) is a newly important invasive pest in many countries in the old world and developing new environmentally friendly management strategies needs a low-cost mass-rearing method for the insect. Here, we designed an economical and effective artificial diet for FAWs based on wheat bran. The rearing performance of this diet was consistent with the traditional artificial diets, but the rearing cost dropped by 26.42% compared to other formulas. This work can promote the mass rearing of FAWs for the application of sterile insect techniques or the production of viral pesticides to control the pest.

## Figures and Tables

**Figure 1 insects-13-01177-f001:**
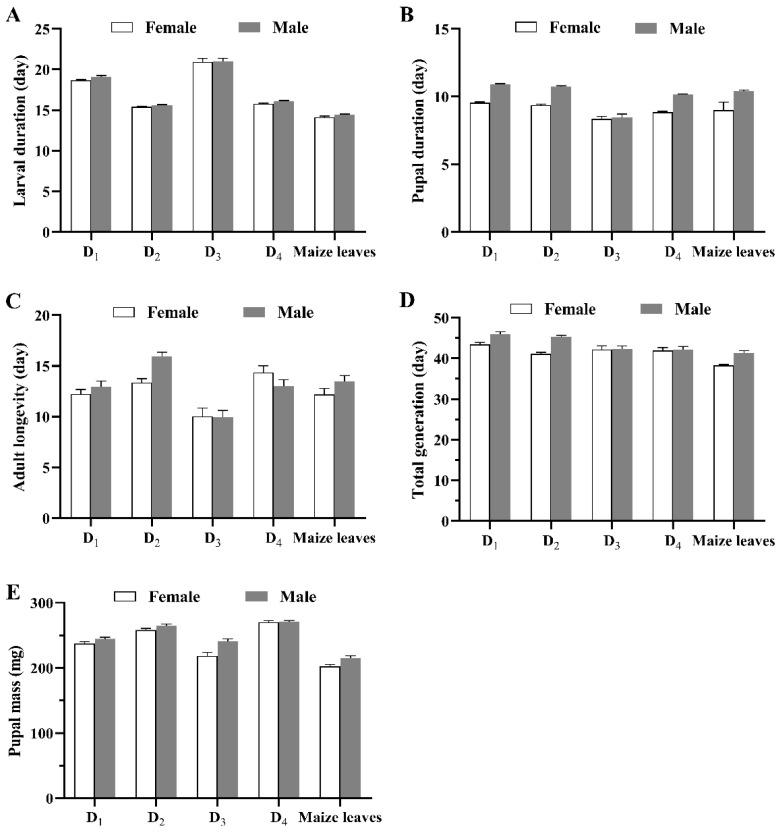
Mean duration (± SE) of developmental stages ((**A**): larval duration; (**B**): pupal duration; (**C**): adult longevity; (**D**): total longevity) and pupal mass (**E**) of female and male *Spodoptera frugiperda* individuals fed different diets.

**Figure 2 insects-13-01177-f002:**
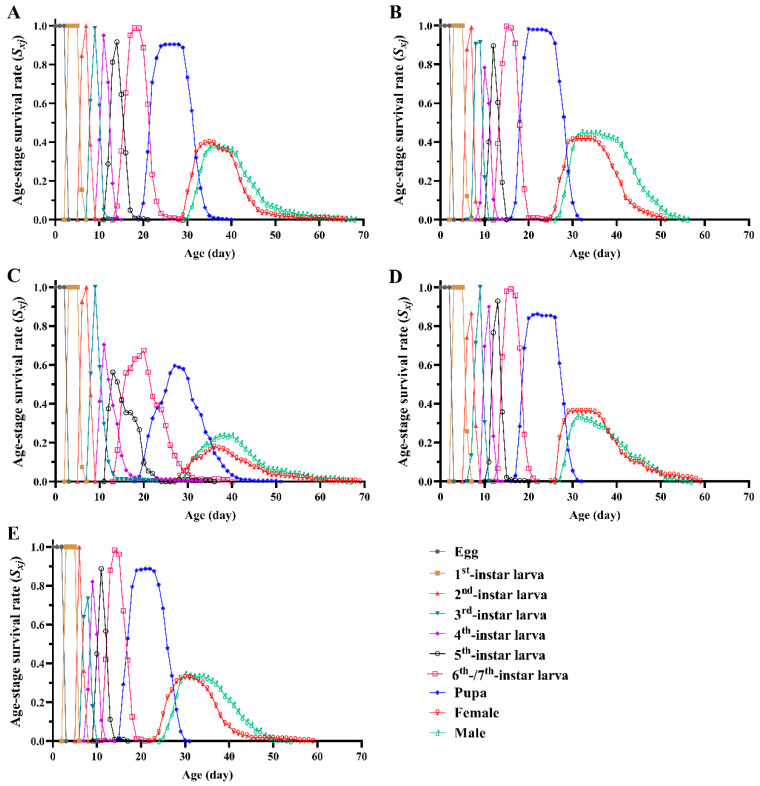
Age-stage survival rate (*s_xj_*) of *Spodoptera frugiperda* individuals fed different diets. (**A**): D_1_; (**B**): D_2_; (**C**): D_3_; (**D**): D_4_; (**E**): maize leaves.

**Figure 3 insects-13-01177-f003:**
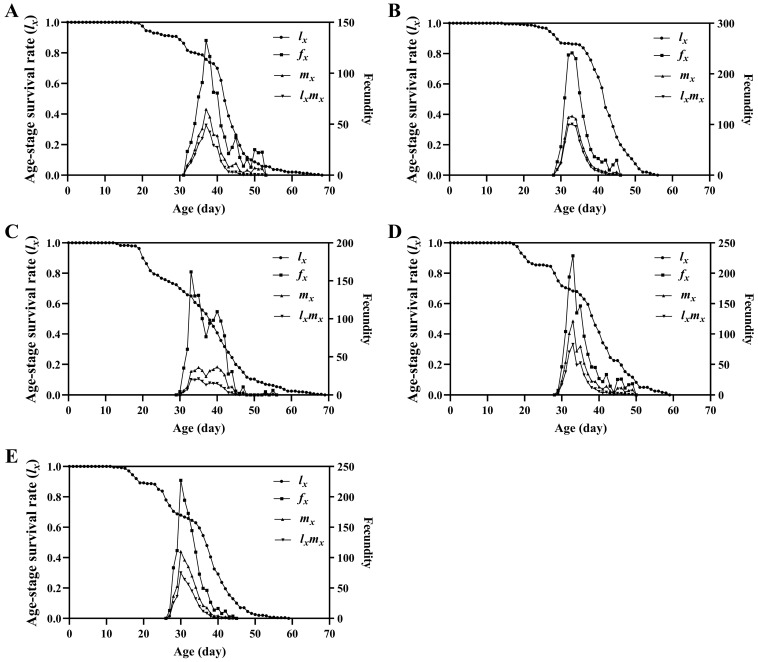
Age-specific survival rate (*l_x_*), age-specific fecundity of female adults (*f_x_*), age-specific fecundity (*m_x_*), and age-specific maternity (*l_x_m_x_*) of *Spodoptera frugiperda* individuals fed different diets. (**A**): D_1_; (**B**): D_2_; (**C**): D_3_; (**D**): D_4_; (**E**): maize leaves.

**Figure 4 insects-13-01177-f004:**
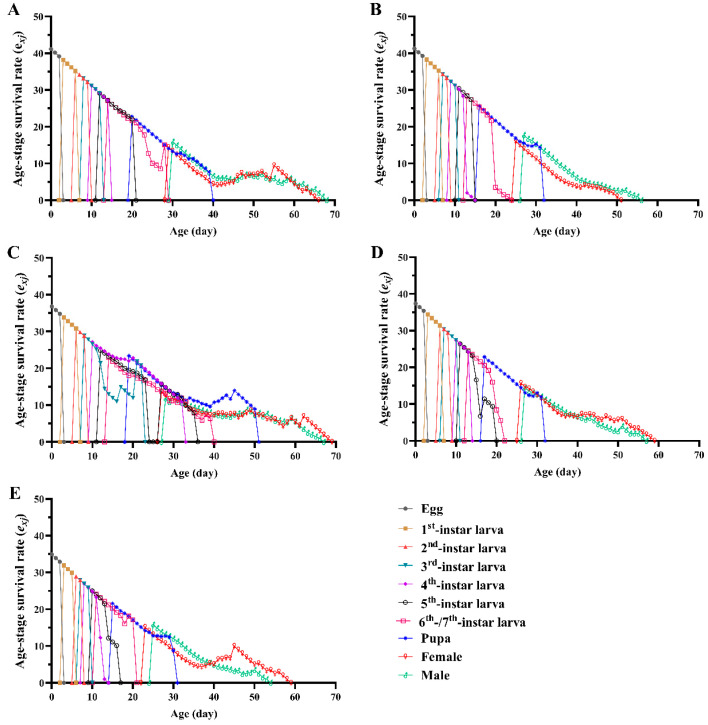
Age-stage specific life expectancy (*e_xj_*) of *Spodoptera frugiperda* individuals fed different diets. (**A**): D_1_; (**B**): D_2_; (**C**): D_3_; (**D**): D_4_; (**E**): maize leaves.

**Table 1 insects-13-01177-t001:** Formulas of four artificial diets fed to fall armyworms (*Spodoptera frugiperda*).

Material	D_1_	D_2_	D_3_	D_4_
Wheat bran (g)	50	150	-	260
Soybean powder (g)	40	80	260	-
Maize powder (g)	100	-	-	-
Yeast powder (g)	30	30	-	-
Agar (g)	24	20	20	20
Casein (g)	40	40	40	40
Sorbic acid (g)	3	3	3	3
Ascorbic acid (g)	3.5	3	3	3
Vitamins (g)	0.15	0.1	0.1	0.1
Formaldehyde (mL)	4	2	2	2
Glacial acetic acid (mL)	4	4	4	4
Distilled water (mL)	1200	1500	1500	1500

The composition of vitamins was as follows: aneurine hydrochloride (7.6%), riboflavin (15.2%), nicotinic acid (30.4%), D-pantothenic acid hemicalcium salt (30.4%), pyridoxine hydrochloride (7.6%), cyanocobalamin (0.6%), folic acid (7.6%) and nicotinamide (0.6%).

**Table 2 insects-13-01177-t002:** Mean duration and reproductive performance (± SE) of developmental stages of FAWs (*Spodoptera frugiperda*) fed different diets.

Parameter	D_1_	D_2_	D_3_	D_4_	Maize Leaves
Egg (day)	3.00 ± 0.00	3.00 ± 0.00	3.00 ± 0.00	3.00 ± 0.00	3.00 ± 0.00
1st-instar larva (day)	3.15 ± 0.02 b	3.12 ± 0.02 bc	3.08 ± 0.02 cd	3.26 ± 0.03 a	3.00 ± 0.00 d
2nd-instar larva (day)	2.25 ± 0.03 b	1.97 ± 0.02 c	2.37 ± 0.03 a	1.90 ± 0.03 c	1.36 ± 0.03 d
3rd-instar larva (day)	2.24 ± 0.03 b	2.04 ± 0.03 c	2.56 ± 0.07 a	2.15 ± 0.03 bc	1.55 ± 0.03 d
4th-instar larva (day)	2.27 ± 0.03 b	1.57 ± 0.03 d	2.75 ± 0.10 a	1.83 ± 0.03 c	1.74 ± 0.03 cd
5th-instar larva (day)	3.08 ± 0.05 b	2.10 ± 0.02 c	3.50 ± 0.09 c	2.18 ± 0.03 c	2.04 ± 0.02 a
6th-/7th-instar larva (day)	5.91 ± 0.08 b	4.73 ± 0.04 c	6.47 ± 0.11 a	4.62 ± 0.04 c	4.50 ± 0.06 c
Larval stage (day)	18.90 ± 0.08 b	15.54 ± 0.06 c	20.94 ± 0.27 a	15.88 ± 0.06 c	14.22 ± 0.08 d
Pupa (day)	10.22 ± 0.07 a	10.08 ± 0.06 ab	9.82 ± 0.08 bc	9.48 ± 0.06 d	9.74 ± 0.07 cd
Egg–pupa (day)	32.10 ± 0.12 b	28.59 ± 0.10 c	33.80 ± 0.33 a	28.40 ± 0.10 c	27.01 ± 0.13 d
Adult longevity (day)	12.60 ± 0.36 b	14.63 ± 0.31 a	9.96 ± 0.50 c	13.70 ± 0.47 ab	12.85 ± 0.42 b
Total generation (day)	44.70 ± 0.39 a	43.28 ± 0.33 ab	42.29 ± 0.55 b	42.10 ± 0.49 b	39.86 ± 0.42 c
Pupal mass (mg)	241.00 ± 1.98 c	261.65 ± 1.90 b	231.48 ± 3.35 d	270.45 ± 1.64 a	209.17 ± 2.22 e
Pre-oviposition period (day)	4.85 ± 0.21 a	4.58 ± 0.21 ab	4.21 ± 0.24 ab	4.71 ± 0.18 ab	3.96 ± 0.16 b
Oviposition period (day)	5.22 ± 0.23 b	6.84 ± 0.21 a	4.90 ± 0.29 b	6.26 ± 0.27 a	6.32 ± 0.23 a
Eggs deposited per female (*n*)	859.69 ± 43.40 b	1439.16 ± 53.71 a	1005.38 ± 77.77 b	1364.78 ± 70.51 a	1261.41 ± 62.72 a
Mating rate (%)	86.57 ± 2.11 ab	93.93 ± 1.92 ab	83.89 ± 2.98 b	89.53 ± 2.13 ab	94.86 ± 1.43 a

Values in the same row followed by different lowercase letters indicate significant differences between different diets (one-way ANOVA, Tukey’s HSD; *p* < 0.05).

**Table 3 insects-13-01177-t003:** Larval and pupal survival rate and adult deformity rate of FAWs (*Spodoptera frugiperda*) fed different diets.

Parameters	D_1_	D_2_	D_3_	D_4_	Maize Leaves
Larval survival rate %	92.08 ± 2.08 ab	98.33 ± 1.10 a	71.25 ± 8.20 b	87.08 ± 4.81 ab	90.83 ± 2.08 ab
Pupal survival rate %	86.44 ± 2.83 a	91.55 ± 2.89 a	89.29 ± 3.63 a	85.15 ± 5.40 a	78.87 ± 2.15 a
adult deformity rate %	0.57 ± 0.56 c	2.37 ± 0.49 bc	14.36 ± 3.48 a	6.05 ± 2.32 ab	5.67 ± 1.91 abc

Values in the same row followed by different lowercase letters indicate a significant difference between different diets (one-way ANOVA, Tukey’s HSD; *p* < 0.05).

**Table 4 insects-13-01177-t004:** Life table parameters (mean ± SE) of *Spodoptera frugiperda* individuals fed different diets.

Parameter	D_1_	D_2_	D_3_	D_4_	Maize Leaves
Net reproductive rate *R*_0_ (offspring/individual)	288.59 ± 29.55 c	580.37 ± 50.13 a	176.53 ± 28.10 d	410.23 ± 44.28 b	384.64 ± 41.97 bc
Mean generation time *T* (d)	37.89 ± 0.27 a	34.25 ± 0.17 b	37.11 ± 0.49 a	34.52 ± 0.22 b	32.16 ± 0.20 c
Intrinsic rate of natural increase *r* (d^−1^)	0.1495 ± 0.0031 c	0.1858 ± 0.0028 a	0.1394 ± 0.0050 c	0.1743 ± 0.0036 b	0.1851 ± 0.0037a
Finite rate of increase λ (d^−1^)	1.161 ± 0.0035 c	1.204 ± 0.0033 a	1.150 ± 0.0057 c	1.1904 ± 0.0043 b	1.203 ± 0.0044 a

Data in same row followed by different letters significantly differed (paired bootstrap test, *p* < 0.05).

**Table 5 insects-13-01177-t005:** Flight performance (mean ± SE) of *Spodoptera frugiperda* individuals fed different diets, recorded in a laboratory using tethered flight mills. Flight performance is expressed using multiple parameters (i.e., distance, duration, and mean velocity).

Parameter	D_1_	D_2_	D_3_	D_4_	Maize Leaves
Flight distance (km)	24.14 ± 1.80 a	22.97 ± 2.19 a	18.88 ± 2.95 b	19.73 ± 1.58 ab	18.71 ± 1.27 ab
Flight duration (h)	7.11 ± 0.39 a	7.12 ± 0.54 a	6.20 ± 0.71 a	6.91 ± 0.46 a	5.65 ± 0.35 a
Flight velocity (km/h)	3.34 ± 0.14 a	3.30 ± 0.18 a	2.67 ± 0.21 b	2.90 ± 0.13 ab	3.44 ± 0.16 a

Different letters in same column indicate significantly different flight parameters (one-way ANOVA, *p* < 0.05, Tukey’s HSD).

**Table 6 insects-13-01177-t006:** Cost per ton of diet material, diet amounts of culturing 1 million FAWs and the cost saving of D_2_ compared with other diets.

Diet	Cost Per Ton (RMB)	Diet Amount of Culturing 1 Million FAWs (ton)	Saving Cost by Culturing 1 Million FAWs (%)
D_1_	6890.00	6.62	43.08
D_2_	5670.00	6.00	23.68
D_3_	5900.00	5.03	12.51
D_4_	3970.00	6.54	-

Exchange rate at the time of writing: RMB 1 = USD 0.1380.

## Data Availability

The data presented in this study are available on request from the corresponding author. The data are not publicly available due to privacy restrictions.

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
