# Peer review of "Wheat-Bran-Based Artificial Diet for Mass Culturing of the Fall Armyworm, Spodoptera frugiperda Smith (Lepidoptera: Noctuidae)"

_insects, 2022, doi:10.3390/insects13121177_

Round 1

Reviewer 1 Report

The manuscript evaluated a cheaper artificial wheat-bran-based diet (D4) for mass rearing the fall armyworm. As a whole, most of the biological parameters of the insect had not been significantly effected with D4 comparing with other diets and the natural food. This work can promote mass rearing the fall armyworm for application of SIT technique or production of viral pesticides to control the insect. There are two problems should be clarified. 

1) In table 1, please give the composition of the "Vitamins";

2) It is not proper to compare the cost of the four diets using diet weight because equal weight does not mean equal efficiency. I suggest the authers use the cost of the diet amounts that can culture the same number of the insect to qualify the saved cost. 

Author Response

Response to Reviewer 1 Comments

The manuscript evaluated a cheaper artificial wheat-bran-based diet (D4) for mass rearing the fall armyworm. As a whole, most of the biological parameters of the insect had not been significantly effected with D4 comparing with other diets and the natural food. This work can promote mass rearing the fall armyworm for application of SIT technique or production of viral pesticides to control the insect. There are two problems should be clarified. 

1) In table 1, please give the composition of the "Vitamins";

Response: Accepted. We have added it in the new version.

The composition of vitamins: aneurine hydrochloride (7.6%), riboflavin (15.2%), nicotinic acid (30.4%), D-pantothenic acid hemicalcium salt (30.4%), pyridoxine hydrochloride (7.6%), cyanocobalamin (0.6%), folic acid (7.6%) and nicotinamide (0.6%).

2) It is not proper to compare the cost of the four diets using diet weight because equal weight does not mean equal efficiency. I suggest the authors use the cost of the diet amounts that can culture the same number of the insect to qualify the saved cost. 

Response: Accepted. We are grateful for your constructive suggestions and comments, and we have adapted it in the Result Section in the new version.

According to the rearing number of FAWs for each diet, the diet amounts of culturing 1 million FAWs for each diet (D1, D2, D3, and D4) were 6.62 t, 6.00 t, 5.03 t, and 6.54 t, respectively. Therefore, compared with D2, using D4 as the artificial diet for rearing 1 million FAWs will save RMB 8056.2 (USD 1111.76) and reduce the cost by nearly 24% (Table 6).

Reviewer 2 Report

Dear Editor:

In reference the manuscript entitled: Wheat-bran-based artificial diet for mass culturing of the fall armyworm, Spodoptera frugiperda Smith (Lepidoptera: Noctuidae)” submitted to the Insect my decision is that work must be accepted with minor changes.

This is a very complete work in which the authors research new formulas for the rearing of Spodotera frugiperda. They found a new formulation with ingredients of low cost, and also maintain the quality and different attributes during the develop and in the adult stage. This information is very important for the future rearing of S. frugiperda, considering that this pest species has high importance in the agriculture trough the world.

Only I found five mistakes and suggestions which in my view point could give more quality the manuscript. These are:

The first part of the abstract is not necessary, I suggest to go directly to the problem. This information could be important the introduction section.

L 86-88 and L 92-93. These comments should be supported by any reference.

L 282…. followed by D4????

L 357 and L 98 Both Anastrepha ludens and Helicoverpa armiguera must be follow by the descriptor. In Anastrepha ludnes (Loew) for example.  

Author Response

Response to Reviewer 2 Comments

In reference the manuscript entitled: Wheat-bran-based artificial diet for mass culturing of the fall armyworm, Spodoptera frugiperda Smith (Lepidoptera: Noctuidae)” submitted to the Insect my decision is that work must be accepted with minor changes.

This is a very complete work in which the authors research new formulas for the rearing of Spodotera frugiperda. They found a new formulation with ingredients of low cost, and also maintain the quality and different attributes during the develop and in the adult stage. This information is very important for the future rearing of S. frugiperda, considering that this pest species has high importance in the agriculture trough the world.

Only I found five mistakes and suggestions which in my view point could give more quality the manuscript. These are:

  • The first part of the abstract is not necessary, I suggest to go directly to the problem. This information could be important the introduction section.

Response: Accepted. We have deleted the sentences in the new version.

  • L 86-88 and L 92-93. These comments should be supported by any reference.

Response: Accepted. We have added some references in the new version.

  • L 282…. followed by D4????

Response: Done.

  • L 357 and L 98 Both Anastrepha ludens and Helicoverpa armiguera must be follow by the descriptor. In Anastrepha ludnes (Loew) for example.

Response: Done. We have added the descriptor in the new version.

Reviewer 3 Report

The search for new ways to fight and control pests and invasive insect species is an urgent task of modern entomology. To develop effective methods of insect control requires a detailed study of their biology. To conduct experiments and create environmentally friendly drugs for pest control, the creation of model insect cultures is required. Therefore, the development of diets for the mass maintenance of insects in laboratory conditions is an urgent task of technical entomology.

The goals and objectives of the work correspond to the main idea of the manuscript. The methods and approaches correctly describe the experiments performed. I have no significant comments on the methodology.

As a recommendation, I ask the authors to make a separate section "5. Conclusion".

Author Response

Response to Reviewer 3 Comments

The search for new ways to fight and control pests and invasive insect species is an urgent task of modern entomology. To develop effective methods of insect control requires a detailed study of their biology. To conduct experiments and create environmentally friendly drugs for pest control, the creation of model insect cultures is required. Therefore, the development of diets for the mass maintenance of insects in laboratory conditions is an urgent task of technical entomology.

The goals and objectives of the work correspond to the main idea of the manuscript. The methods and approaches correctly describe the experiments performed. I have no significant comments on the methodology.

As a recommendation, I ask the authors to make a separate section "5. Conclusion".

Response: Accepted. We have added Conclusion Section in the new version.

Spodoptera frugiperda (FAW) is a newly important invasive pest in many countries of the old world, and developing new environmentally friendly management strategies needs a low-cost mass rearing method of the insect Here, we designed an economical and effective artificial diet of FAW based on wheat bran, the rearing performance of this diet was consistent with the traditional artificial diets, but the rearing cost was dropped nearly 26.42% in comparison with other formulas. This work can promote mass rearing FAW for application of sterile insect technique or production of viral pesticides to control the pest.